# Automatic detection of pupil reactions in cataract surgery videos

Natalia Sokolova[1]*, Klaus Schoeffmann[1], Mario Taschwer[1], Stephanie Sarny[2,3], Doris Putzgruber-Adamitsch[2], Yosuf El-Shabrawi[2,3]*

1 Institute of Information Technology, Klagenfurt University, Klagenfurt, Austria, 2 Department of Ophthalmology, Klinikum Klagenfurt, Klagenfurt, Austria, 3 Department of Ophthalmology, Medical University Graz, Graz, Austria

* natalia@itec.aau.at (NS); yosuf.elshabrawi@medunigraz.at (YES)

**Data Availability Statement:** All relevant data are within the manuscript and its Supporting information files.

## Abstract

In the light of an increased use of premium intraocular lenses (IOL), such as EDOF IOLs, multifocal IOLs or toric IOLs even minor intraoperative complications such as decentrations or an IOL tilt, will hamper the visual performance of these IOLs. Thus, the post-operative analysis of cataract surgeries to detect even minor intraoperative deviations that might explain a lack of a post-operative success becomes more and more important. Up-to-now surgical videos are evaluated by just looking at a very limited number of intraoperative data sets, or as done in studies evaluating the pupil changes that occur during surgeries, in a small number intraoperative picture only. A continuous measurement of pupil changes over the whole surgery, that would achieve clinically more relevant data, has not yet been described. Therefore, the automatic retrieval of such events may be a great support for a post-operative analysis. This would be especially true if large data files could be evaluated automatically. In this work, we automatically detect pupil reactions in cataract surgery videos. We employ a Mask R-CNN architecture as a segmentation algorithm to segment the pupil and iris with pixel-based accuracy and then track their sizes across the entire video. We can detect pupil reactions with a harmonic mean (H) of Recall, Precision, and Ground Truth Coverage Rate (GTCR) of 60.9% and average prediction length (PL) of 18.93 seconds. However, we consider the best configuration for practical use the one with the H value of 59.4% and PL of 10.2 seconds, which is much shorter. We further investigate the generalization ability of this method on a slightly different dataset without retraining the model. In this evaluation, we achieve the H value of 49.3% with the PL of 18.15 seconds.

# 1 Introduction

Nowadays, content-based analysis of surgical videos plays an important role in the area of medical research, where it often helps to improve the quality of these surgeries. One example is cataract surgery, the most commonly performed surgery worldwide. During this surgery, the natural lens of the human eye is replaced by an artificial intraocular lens. With the increased needs of our societies in regards to their visual abilities, the use of so-called premium

**Funding:** This work was funded by the FWF Austrian Science Fund under grant P 31486-N31.

**Competing interests:** No authors have competing interests.

intraocular lenses (IOL), such as EDOF IOLs, multifocal IOLs or toric IOLs becomes more and more established. However, due the optic behavior of these lenses even minor intraoperative deviations may lead to a significant sometimes even permanent visual loss. In multifocal IOLs, for instance even minor decentrations or an IOL tilt, that would be optically insignificant in conventional monofocal IOLs can hamper the visual performance of these IOLs. In toric IOLs a perfect alignment of the torus is of utmost importance, as otherwise the functionality of the IOL is reduced. A perfect alignment or centration is often hard to achieve in eyes, where intraoperatively a contraction (miosis) of the pupil occurs.

To detect events that might lead to these pupillary changes, or just be able to measure the incidence that pupillary changes occur or to what extend these chance are seen will most likely improve the outcome of cataract surgeries. Therefore, a post-operative analysis of large data sets of cataract surgeries is needed, especially if these data files could be evaluated automatically. However, the video collection of cataract surgeries at every hospital is huge and increases on a daily basis. Up-to-now surgical videos are evaluated mostly by looking at a limited number of surgeries only, or in case of pupil reactions, by measuring pupil chances in a small number of intraoperatively taken pictures [1]. These tasks are especially time-intensive and the clinical relevance of results drawn are somewhat limited. On the other hand, if the clinician had the ability to find as many pupil reactions as possible in a large collection of surgeries for statistical analysis, the results of a study evaluating intraoperative variation of pupil size and its clinical outcome might improve significantly. Thus, we aim to automatically detect pupil reactions in cataract surgery videos, which—to the best of our knowledge—has not been done before.

We already proposed an approach for pupil and iris segmentation in video frames of cataract surgery and evaluated the achievable performance in [2]. In this work, we employ the proposed method to track the size of the iris and the pupil throughout the entire recording and localize the pupil reactions. For this purpose, we compare two different approaches: (1) localization and tracking of the pupil, and monitoring its width size over time, (2) localization and tracking of the pupil and the limbus width, and monitoring the difference of them over time. The first approach is based on the assumption that the limbus is constant during pupil reactions, while the pupil reduces or increases its size. In order to evaluate the achievable accuracy of both approaches, we collect and annotate a dataset of 10 cataract surgery videos (118,413 frames) with pupil reactions of different sizes. To evaluate our approach, we bring in a *Ground Truth Coverage Rate (GTCR)* metric, which shows, how many correct predictions correspond to a single correctly detected ground truth. We are able to achieve an H value of 60.9%, which is a harmonic mean of *recall*, *precision* and *GTCR*, and 18.93 seconds of average prediction length (PL). However, we recommend to use the configuration with the H value of 59.4%, which has a PL value of 10.2 seconds, which is more appropriate for the clinicians, as the average pupil reaction duration is one second. Moreover, the GTCR of retrieving only medium and strong changes using the best configuration is around 100%, while the *recall* of detecting such changes is very high. It reaches 90.60% for medium and strong pupil reactions.

In addition, we evaluate the generalization performance of our approach. For this purpose, we collect and annotate a different dataset, which was recorded at the same hospital with a different camera. According to our results, the detection performance does not degrade too much for this additional dataset with significantly different content characteristics and still achieves an H value of 49.3%. Also, the PL value on this dataset is 18.15 seconds, which is very similar to the previous one (18.53 seconds).

The rest of the paper is organized as follows. First, Section 2 discusses the existing related research approaches. Section 3 describes the datasets used for parameter optimization and generalization evaluation. Then, Section 4 presents pupil and limbus width tracking

methodology. Section 5 describes the used metrics, obtained results for both datasets, and their comparison the expert opinions. Finally, Section 6 summarizes the main paper's conclusions.

## 2 Related work

The topics of iris and pupil localization and analysis, as well as eye-tracking, have been studied a lot in the research community within the last few years. These methods have been applied in various fields. One of the areas where iris analysis is explored a lot and even used afterward in daily life is biometric identification. For example, in [3, 4] authors segment the iris for further personality identification.

In the field of ophthalmology, McConnon et al. [5] evaluate the effect of different ophthalmological diseases on iris segmentation. In [6–8], the authors apply a similar approach to diagnose diseases, such as pterygium, strabismus, and cataract. Lakra et al. [9] compare the eyes before and after cataract surgery. Additionally, the authors of [10] introduce an eye gaze tracking control interface for the surgical system based on iris segmentation.

In some other works, researchers already apply pupil and iris segmentation methods in cataract surgery videos. For example, Ektesabi et al. [11] define at first the personal color characteristics for every patient and then use color histograms to localize pupil and iris to use it as a prepossessing step for further analysis. In [12, 13], the authors localize iris and pupil for further instrument localization and classification. The images are transformed into the YUV color space, and then a binary mask is obtained with an assumption that the color of the pupil is different from everything else in the frame. From this mask, the most probable pupil circle is retrieved using Hough transform [14]. To decrease the fault rate, the authors propose to use pixel and circle centers counting. A similar approach is presented as a preprocessing step by Quellec et al. in [15, 16]. They use Canny edge detection and Hough transform to find the center of the pupil for eye tracking to detect the region of interest. In [17], the authors first use a random sample consensus algorithm to detect the pupil in the image to find the most probable circles of the pupil to combine them afterward into an elliptical fit. Second, they search for the circle outside of the pupil, which has the highest gradient value between inner and outer areas to detect the iris.

The importance of pupil size alterations is also proved in such medical researches as [18–20]. The authors of these works explore the effect of pupil size changes on cataract surgery and evaluate different medications that help to achieve the adequate pupil size within the entire surgery. The pupil diameter measurements in femtosecond laser-assisted cataract surgery frames are also done in [21, 22].

To the best of our knowledge, automatic pupil reaction detection has not been addressed yet.

## 3 Materials

A written informed consent was obtained from the subjects. No minors participated in this research work. The consent form and this study were approved by the local ethics committee. The Carinthian ethics committee granted approval under the Ek number A28/17. The videos used in this study are fully anonymized.

### 3.1 Dataset A

For parameter optimization, we collected a dataset of 10 cataract surgery videos (118,413 frames) that we randomly selected from around 600 videos recorded at Klinikum Klagenfurt in 2017–2018. The videos have the following parameters:

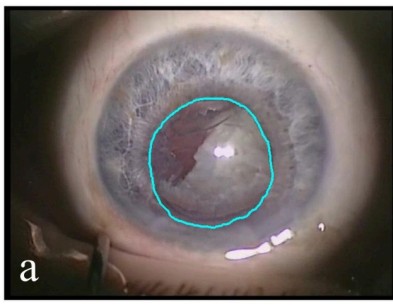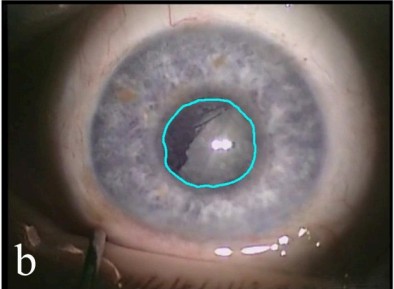

**Fig 1. Example of pupil reaction from Dataset A.** The cyan line shows the pupil segment, which significantly changes its size during the operation. A) The eye before pupil reaction. B) The eye after pupil reaction.

- a resolution of 540x720 pixels;

- a frame rate of 25 fps.

In all these videos, we manually annotated pupil reactions—which are size changes of the pupil that may suddenly happen during surgery (see Fig 1). All of these reactions are split into three categories in terms of their size: strong, medium, and weak. In Fig 1 you can see an example of a pupil (a) before and (b) after strong pupil reaction. The videos in the dataset contain a total number of 135 pupil reactions. The amount of weak, medium and strong pupil size changes is 72.6%, 14.8%, and 12.6%, respectively. The average duration of a pupil reaction is 1 second, with a standard deviation of 0.67 second.

In our annotations, we add a flag to every pupil reaction to note, if it is difficult to detect or not. For example, the surgeon can push the eye away from the highlighted area, so only a small part of the eye is illuminated. If the pupil reaction occurs at this time, or, for example, if it happens very slightly, it is barely visible even for clinicians. Such changes we denote as *unclear*. In total, 50% weak, 0% medium, and 5.9% strong changes are unclear. All other changes are called *obvious* in the remainder of the paper.

## 3.2 Dataset B

In order to evaluate the generalization performance of our model, we collect another dataset that has better visual quality. This dataset consists of videos from 20 cataract surgeries (532,477 frames) that were randomly selected from around 650 videos recorded at Klinikum Klagenfurt in 2019–2020. The videos have the following parameters:

- a resolution of 1280x720 pixels;

- a frame rate of 60 fps.

The visual quality, as mentioned above, is slightly better, because the resolution and frame rate are higher. This conditions let partially eliminate the motion blur effect. However, it is impossible to avoid the blur completely because, as mentioned already, the microscope needs to be adjusted at the beginning of every operation for the second eyepiece (which is the basis for the video recording). If the setting of the microscope is changed during the operation, for example, because the patient moved, it could be required to adjust the second eyepiece again. If this is not correctly performed by a second person (e.g., a clinical assistant), the video recording's visual quality would not be optimal. Also, the aspect ratio of those videos is different (16:9 instead of 4:3 for the videos in the previous Dataset A). The total amount of pupil

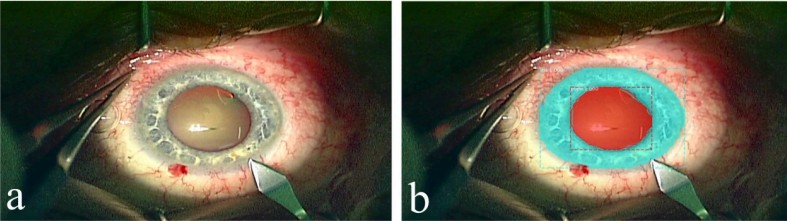

**Fig 2. Original (A) and automatically segmented (B) frames from the Dataset B.**

reactions in Dataset B is 229. The amount of weak, medium and strong changes is 77.3%, 17.9%, and 4.8%, accordingly. All of annotations are also accompanied by a flag, which shows if the pupil reaction is obvious or unclear.

Example frames from Dataset B with and without automatic segmentation of pupil and iris are presented in Fig 2.

## 4 Methodology

In this work, we operate with the following terms: pupil, iris, and limbus (a line between the iris and the outer part of the eye). All of them are presented in Fig 3.

As mentioned above, we need first to localize the pupil and iris in every video frame. For this purpose, we use the same method as already described in [2] and a dataset consiststing of 82 frames from 35 cataract surgery videos, collected at Klinikum Klagenfurt, Austria.

### 4.1 Pupil and iris segmentation

To implement this task, we use state-of-the-art pixel-accurate segmentation provided by the Mask R-CNN network [23]. It has proven as an efficient tool for medical analysis in the medical domain, as research works for surgical tool localization and classification in cataract [24] and laparoscopy [25] surgery videos show. It is also known to work well for object segmentation in the general domain. In our evaluations, the following Mask R-CNN training configuration was used:

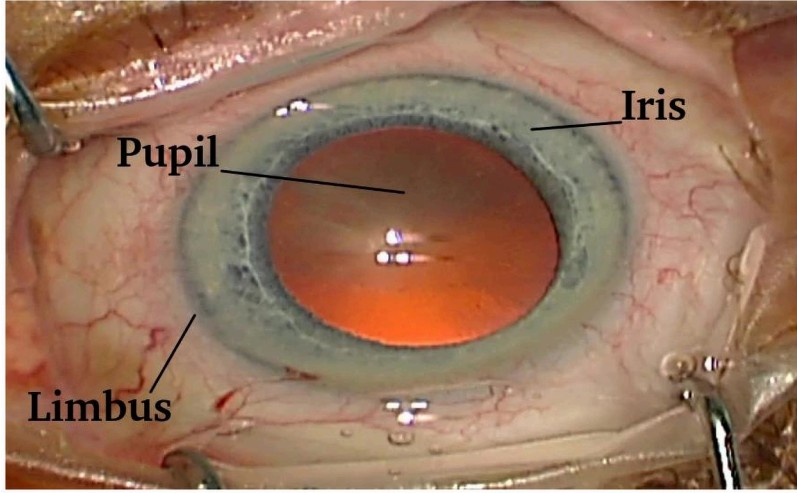

**Fig 3. Structure of the human eye.**

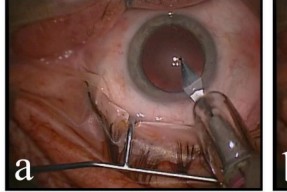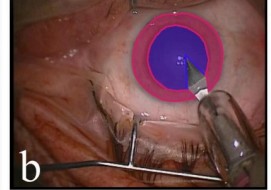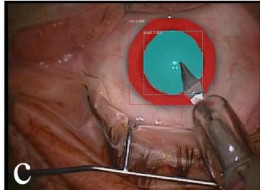

**Fig 4. (A) Original, (B) annotated, and (C) automatically segmented frames.**

- ResNet-101 as a backbone network [26];

- learning rate of 0.002;

- momentum of 0.9;

- batch size of 1;

- 100 epochs;

- 5 validation steps.

We use the Mask R-CNN implementation provided by [27] with Tensorflow as a backend and an Nvidia Titan RTX N GPU with 24 GB of RAM. The backbone network (ResNet-101) is pre-trained with the Microsoft COCO (Common Objects in Context) dataset [28] and these weights are used further during training.

As we have shown in [2], this configuration is able to obtain high performance for pupil and iris segmentation. It achieves 100% IoU (Intersection over Union) for at least 90% of pupil and 75% of iris segments. The IoU metric describes the ratio of the intersection of the ground truth segment with the predicted segment divided by the union of both of them. The trained Mask R-CNN model is then applied to Dataset A and Dataset B and automatically predicts two segments in each frame—one for the iris and another one for the pupil. Fig 4 shows an exemplary frame, its ground truth annotation, and the automatically obtained pupil and iris segments.

## 4.2 Temporal interpolation and size measurement

Once we are able to segment the pupil and iris in every frame, we perform temporal interpolation and measure the size of the pupil and the outer iris border (in the remainder of the paper will be denoted by limbus). Both values are tracked over the video in order to detect pupil reactions (see Fig 5).

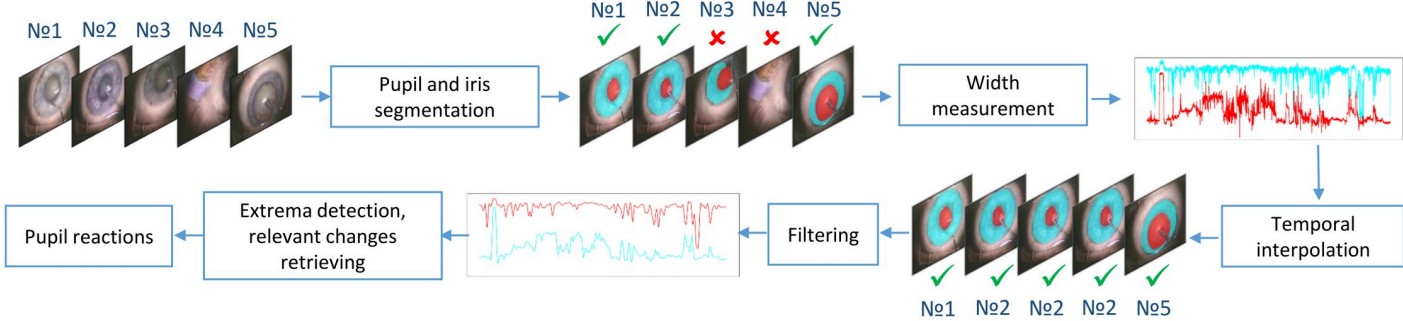

**Fig 5. Overview of the proposed pupil reaction detection approach.**

For further processing, we only use frames where both, pupil and iris, are presented in the frame. Therefore, we set the limit of the predictions per each frame and the number of predicted classes to 2. Then, for each frame, we compute a boundary box from the predicted masks, which is more precise than the one that the network provides [2]. Here, we only take into account frames where the boundary box of the pupil is fully contained within the iris boundary box.

During the operation, the eye moves vertically much more often than horizontally, thus measuring the width of the boundary box is more stable and gives more robust results. As a result of this procedure, we get a list of frame numbers associated with the pupil and limbus width values. To keep the numbering of the frames, we insert the missing values with the last available pupil and iris width values. When we have missing values at the beginning of the video, we fill them with the next following width values that satisfy our conditions. This allows us to get rid of missing values and to align the width measurements with the actual time line of the surgery video. These steps are presented in Fig 5 as *temporal interpolation* and *width measurement*. For simplicity, the resulting measurements of pupil and limbus width are named signals in the remainder of the paper.

## 4.3 Low-pass filtering

As apparent in Fig 5, the automatically detected sizes of the pupil (red) and limbus, which is the outer border of iris (cyan), slightly oscillate over time, what complicates the automatic detection of pupil reactions. To get rid of this problem, we apply a low-pass filter to both signals in order to smooth them. We use the Butterworth low-pass filter (https://docs.scipy.org/doc/scipy/reference/generated/scipy.signal.butter.html) from the scipy python library to clean the signals with two-fold signal processing (https://docs.scipy.org/doc/scipy/reference/generated/scipy.signal.filtfilt.html) to eliminate the phase shift. For our experiments we use the following cutoff frequency ($f$) values: {0.01, 0.05, 0.1, 0.15, 0.2} Hz. An exemplary result can be seen in Fig 5 after the *filtering* step.

Also, a more detailed example of filtered pupil (red) and limbus (cyan) values is presented in Fig 6, together with filtered signals (black) at a cutoff frequency of 0.1 Hz. It is visible in the figure, that filtering significantly smoothens the signals, which helps to recognize the real pupil reactions automatically. The pictures on top of the graph represent the predictions in the cataract surgery frames at corresponding times.

## 4.4 Relevant pupil size changes retrieving

To retrieve the pupil reactions, we focus on the extreme values in the filtered signals. In particular, we consider every signal segment between two extreme neighbor points as a potential pupil reaction. Its size is denoted by the difference of signal values at the extreme points at the beginning and the end of the detected signal segment. To determine the pupil reaction length, we compute the difference between frame numbers at these extreme points.

The pupil may react multiple times within the operation, as you can notice from the Fig 6, but it has a prevailing size among the entire surgery. Usually, the pupil returns to this state during phases where no liquid is inserted through the surgical tools, or when there are no instruments in the eye. At the same time, the limbus size does not change within the entire surgery.

However, although the camera zoom usually remains quite stable during cataract surgery, we track limbus and pupil independently from each other in order to get rid of wrong detections. More precisely, we evaluate the difference between these two signals, since the limbus stays quite constant during actual pupil reactions, while the pupil size either increases or decreases. We assume, that such method might give better results.

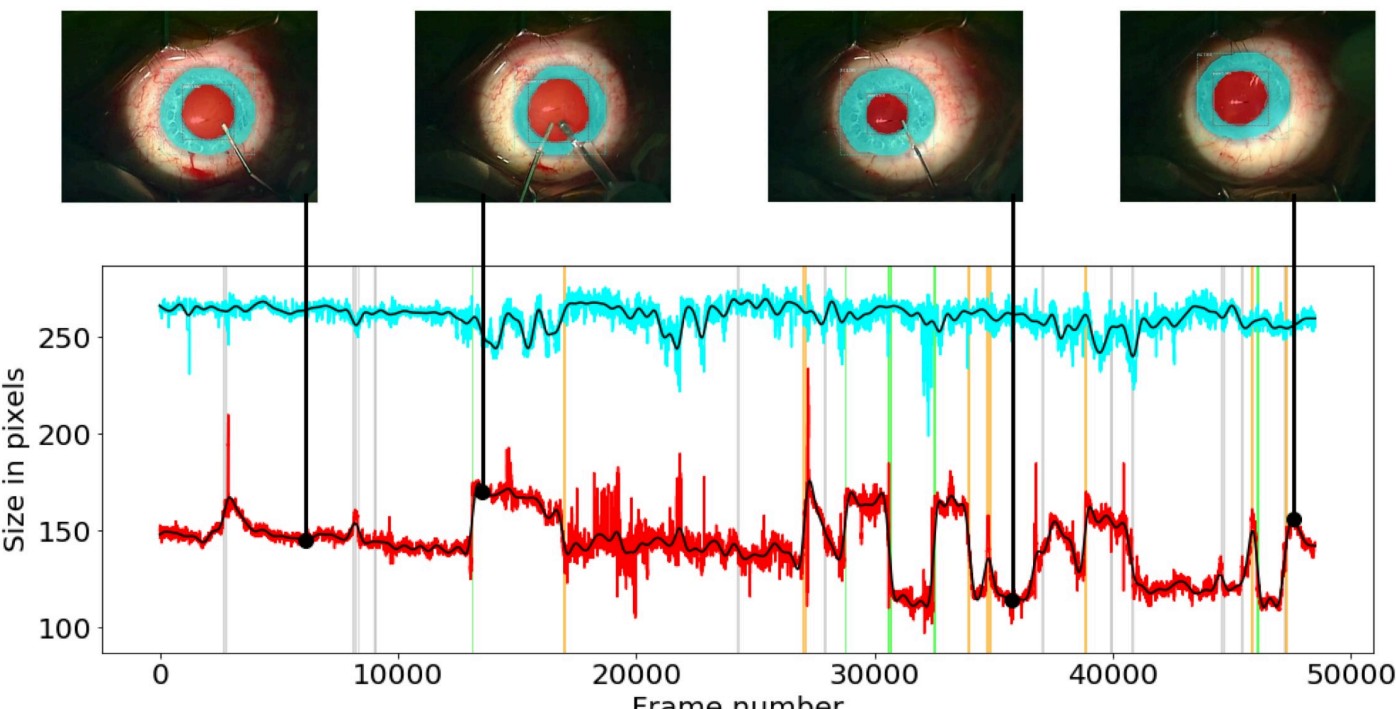

**Fig 6. Example of the original limbus (cyan) and pupil (red) width values.** Both signals are filtered with the Butterworth filter and a cutoff frequency of 0.1 Hz (black) in one video with strong (green), medium (orange), and weak (gray) annotations. The pictures over the graph are the automatically segmented frames at corresponding times.

To filter out the irrelevant pupil reactions in terms of the size, we use the difference between median values of limbus and pupil widths (*DM*) as a reference value. In particular, we consider as relevant all pupil reactions where the pupil size change is not less than {0, 0.02, 0.04, 0.06, 0.08, 0.1, 0.12, 0.14, 0.16, 0.18, 0.2} of the *DM*. In the remainder of the paper, we denote such *DM* thresholds as *Delta*.

The symbols *f* and *Delta* are the hyper parameters that need to be optimized for the dataset at hand.

## 5 Experimental results and discussion

### 5.1 Evaluation metrics

We evaluate the performance of our pupil reaction detection method using Recall, Precision, and Ground-truth Coverage Rate (GTCR), which are defined as follows:

$$Recall = \frac{tp}{tp + fn} \tag{1}$$

$$Precision = \frac{tp}{tp + fp} \tag{2}$$

In the equations above, *tp*—true positive—is the amount of all correctly detected ground truth segments (i.e., correctly detected pupil reactions). *Fn*—false negative—is the amount of all ground truth segments which were not found. *Fp*—false positive—is the number of all

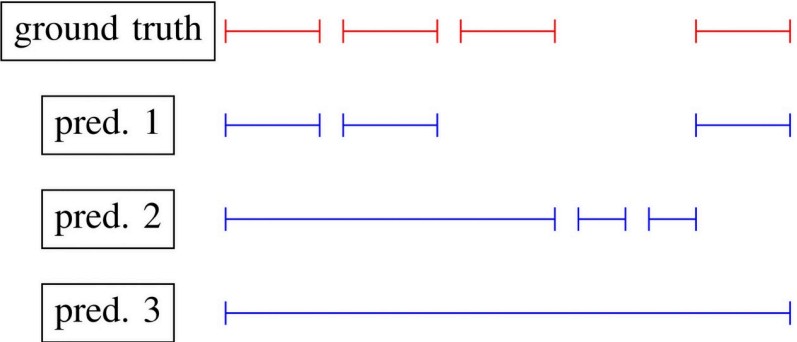

**Fig 7. Example of three possible predictions (blue) in accordance with the existing ground truth (red).** Every line segment represents a pupil reaction.

predicted segments that *do not contain any* correctly predicted ground truth segment.

$$GTCR = \frac{P_c}{tp} \tag{3}$$

In Eq 3, $P_c$ is the number of predicted segments, with which at least one ground truth was detected correctly (correct prediction). The Ground Truth Coverage Rate (GTCR) represents how many correct predictions correspond to one correctly detected ground truth segment ($tp$).

In Fig 7, you can see an example of annotated pupil changes (red) and three possible predicted sets of pupil changes (blue). For the first situation (*pred.* 1), recall, precision and Ground Truth Coverage Rate (GTCR) are 75% (3 out of 4 ground truth segments are found), 100% (there are no wrong predictions), and 100% (in average, 1 detected ground truth is found by 1 correct prediction), respectively.

For *pred.* 2, these metrics are 75%, 33% (two out of three predictions are wrong), and 33% (in average, three detected ground truth segments are found by one correct prediction) for recall, precision, and GTCR, respectively. At the same time, for *pred.* 3 they are 100%, 100%, and 25%. The recall and precision values for *pred.* 3 are very high, but it is noticeable that the predicted segment is not as good as we expect. In this case, we can see that the GTCR metric is important, and it shows how exact the found pupil size changes are segmented, and how reliable the values of recall and precision are.

To find the best configuration of parameters we evaluate the harmonic mean (H) value of *Recall*, *Precision*, and *GTCR* (Eq 4). For the examples in Fig 7, H is 90% for *pred.* 1, 40.6% for *pred.* 2, and 50% for *pred.* 3.

$$H = \frac{3}{\frac{1}{Recall} + \frac{1}{Precision} + \frac{1}{GTCR}} \tag{4}$$

Although, the H value represents the performance of our method in a nice way, it does not take into account the duration of the predicted segments. To analyze this parameter, we compute the average duration of all predicted segments for each configuration (Fig 8). Average duration of pupil reaction in is around one second. For both pupil (Fig 8A) and difference (Fig 8B) signals the average duration for *f* of 0.01 Hz is unacceptably high. In particularly, the duration of predicted video fragments is longer than one minute in this situation. As it is too long in comparison with the average ground truth duration, we exclude it from the graphs and we

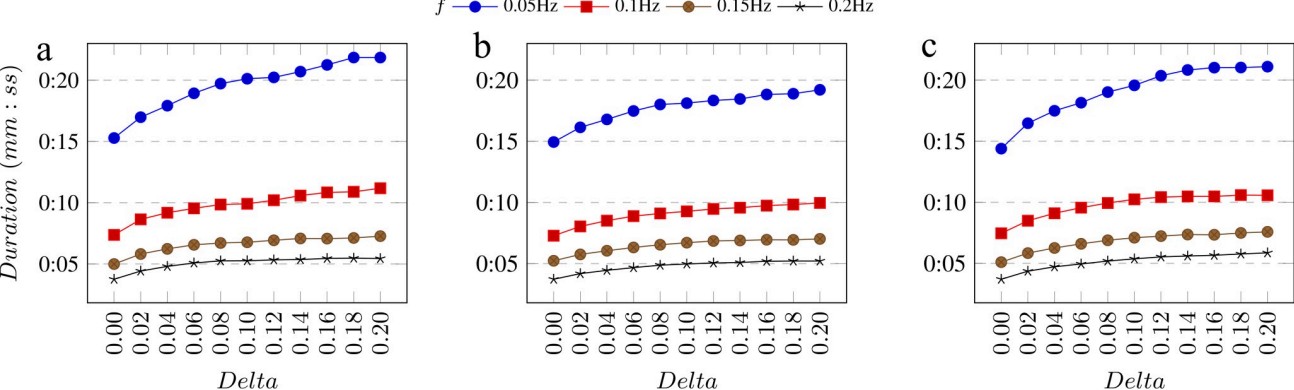

**Fig 8. Average predicted segment duration.** (A) For pupil width signal and (B) for difference signal on dataset A, (C) for pupil width signal on dataset B with different *f* and *Delta*.

do not consider the configurations with this *f* value as the best ones, even if they obtain better H values. In addition, average prediction length for *f* of 0.05 Hz is also quite long (15 up to 22 seconds), therefore we do not use any lower *f* values for our evaluations.

To get rid of too long predicted segments we introduce an additional threshold, based on the IoU value for ground truth and predicted segments. We include in our evaluation only those predicted segments whose IoU with any ground truth segment is greater than the following number:

$$IoU_{tr} = \frac{min(GTL, PL)}{max(GTL, PL)} \tag{5}$$

where PL—average prediction length, GTL—average ground truth length.

## 5.2 Evaluation results

To evaluate the performance of our algorithm, we use, as mentioned, H value (harmonic mean) to choose the best hyper parameters configuration, since *Recall*, *Precision*, and *Precision* diverge a lot.

**5.2.1 Parameter optimization.** First, we compare the ability of our method to detect all types of pupil reactions in terms of size and difficulty on pupil and difference signals. The results of these evaluations are presented in Fig 9A and 9B. We can see that the best configuration for pupil signal (*Method A*) was achieved with *f* of 0.05 Hz and *Delta* of 0.06, resulting in an H value of 60.9%.

At the same time, the best H value for difference signal (*Method B*) is 59.4% with the *f* of 0.05 Hz and *Delta* of 0.1. Based on that, we conclude that our method achieves for the pupil signal almost the same (though slightly better) results as for the difference signal.

We explain this result with the fact, that the eye moves quite intensively within the surgery, which can affect the segmentation precision of the iris more than of the pupil and create significant additional noise. In particular, the iris may go behind the highlighted area or even the frame border more frequently and further that pupil.

In the next place, we evaluate the *Method A* on only obvious pupil reactions (*Method C*). As unclear changes are difficult to detect even with the human eye, we assume that the results should be better if we exclude such pupil size changes from our evaluation. However, according to Fig 9C, the best result for this configuration is achieved with *f* of 0.05 Hz and *Delta* of

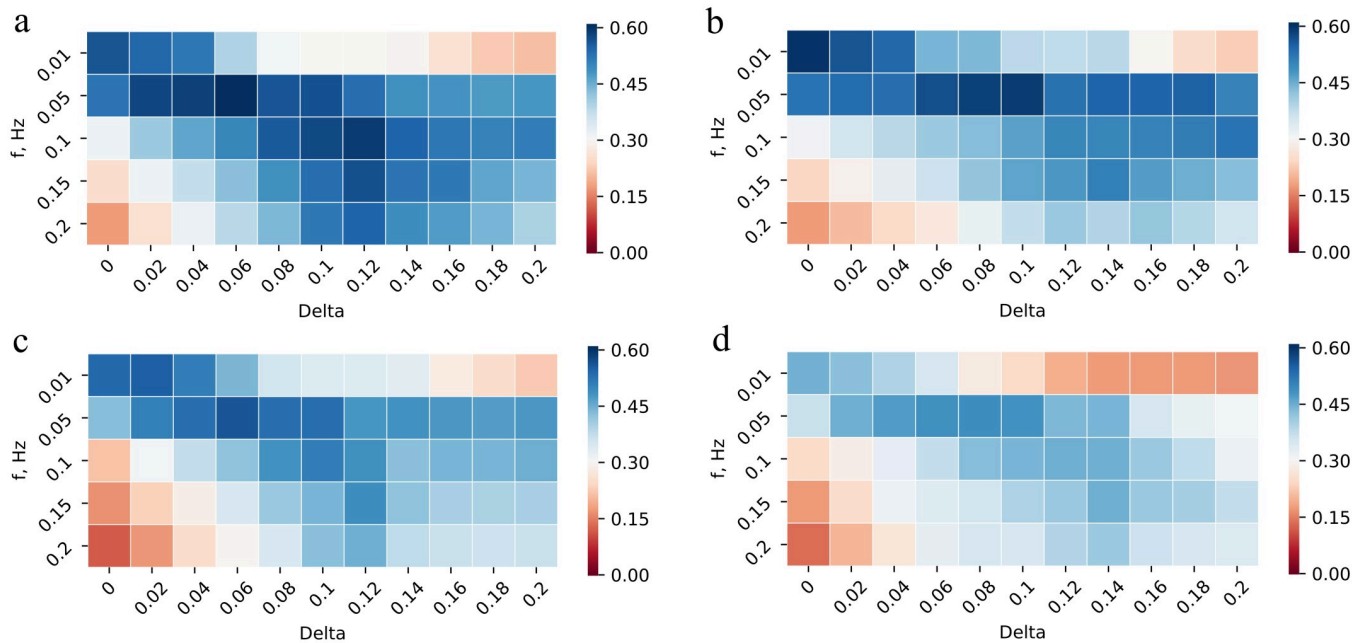

**Fig 9. H value for different methods.** A) Method A: all types of pupil changes in terms of size and difficulty on the pupil signal on Dataset A. B) Method B: all types of pupil changes in terms of size and difficulty on the difference signal on Dataset A. C) Method C: obvious pupil changes of any size on the pupil signal on Dataset A. D) Method D: all types of pupil changes in terms of size and difficulty on the pupil signal on Dataset B.

0.06, resulting in an H value of 56.4%. It means that our assumption was wrong. Therefore, we consider as the best methodology *Method A*. Better performance of this configuration could be explained because a person needs to see the pupil reaction, while our method can compare the pupil size before and after the pupil reaction happens outside the visible area.

**5.2.2 Predicted segment duration.** For *Method A*, a PL (average prediction length) value for the best configuration (I) is 18.93 seconds with an *f* value of 0.05 Hz and *Delta* of 0.06. While for *f* of 0.1 Hz and *Delta* of 0.12 (configuration II) the PL value is 10.2 seconds and the H value is 59.4%, which is a second highest number for this method (see Fig 9A). This value is also higher than any H value of *Method B* or *Method C*. For both mentioned configurations, we analyze the recall, precision and GTCR in detail. The results are presented in Table 1. The parameters of these configurations are quite close to each other, while PL of configuration II is much shorter. Therefore, we consider this configuration as the best one for practical use.

**5.2.3 Performance evaluation for pupil reactions of different sizes.** Clinicians need to detect as many pupil reactions as possible, but the stronger the reactions are, the more significant they are. Thus, for the best and recommended detected configurations (I and II), we closely look at the detection performance for pupil reactions of different sizes. Since annotated

**Table 1. Detailed results of the best configurations for *Method A* and *Method D*.** Presented metrics (*H, Recall*, etc.) are described in Section 5.1. The configurations parameters are: Configuration I (*Method A*, best configuration) has *f* = 0.05, *Delta* = 0.06, Configuration II (*Method A*, recommended configuration) has *f* = 0.1, *Delta* = 0.12, Configuration III (*Method D*) has *f* = 0.05, *Delta* = 0.08.

| Configuration | H | Recall | Precision | GTCR | PL |
|---|---|---|---|---|---|
| I | 60.94% | 64.31% | 45.53% | 85.35% | 18.93s |
| II | 59.44% | 66.68% | 41.78% | 86.69% | 10.2s |
| III | 49.32% | 41.38% | 43.25% | 73.83% | 18.15s |

**Table 2. Detailed results of the configurations I and II for pupil reactions of different intensities.**

| Reaction intensity | H | Recall | Precision | GTCR |
|---|---|---|---|---|
| | Configuration I | | | |
| All | 60.94% | 64.31% | 45.53% | 85.35% |
| Medium and strong | 52.17% | **90.64%** | 27.53% | **98.57%** |
| Strong | 35.87% | **93.94%** | 15.88% | **100%** |
| | Configuration II | | | |
| All | 59.44% | 66.68% | 41.78% | 86.69% |
| Medium and strong | 45.96% | **82.99%** | 23.14% | **100%** |
| Strong | 35.44% | **100%** | 15.47% | **100%** |

pupil reactions are split into three groups (weak, medium, and strong), we repeat the evaluation for these groups. The first group includes pupil reactions of all sizes; the second group contains only medium and strong pupil reactions, while the third group consists of strong pupil reactions only.

According to Table 2, *Recall* and *GTCR* values increase accordingly with the intensity of pupil reactions, while *Precision* simultaneously decreases. Even though the *Precision* is quite low for second and third groups of pupil reactions, their *Recall* is very high. For example, for configurations I and II, it is more than 90% and 80% for both groups, respectively. It means that our algorithm returns most of the relevant results that exist in the processed videos. As well as *Recall*, the GTCR value of detecting strong and medium reactions is very high and reaches 98.57% and 100% for configurations I and II, respectively. Meanwhile, both configurations achieve 100% GTCR for retrieving only strong changes. It means that the algorithm cannot ideally segment the weak reactions, but it detects strong and medium pupil size changes very accurately. In other words, for strong reactions for example, each correctly detected by the algorithm ground truth was found with only one prediction segment (in contrast to longer predictions that would include several correctly detected ground truth segments).

## 5.3 Generalization results

As already mentioned, we evaluate our methodology (*Method A*) on a slightly different dataset B (*Method D*). It turned out that the best configuration for the new dataset is obtained with the *f* of 0.05 Hz and *Delta* of 0.08 (configuration III). As you can see in Fig 9D, the H value does not degrade too much for this dataset. We achieve for these videos H value of 49.3%. At the same time, the PL is almost the same (18.15 vs 18.53 seconds for dataset A). The detailed results for this configuration are shown in Table 1. In this case, recall value degrades the most. This could happen because the amount of strong pupil reactions in dataset B is less than in dataset A (4.9% vs 12.6%), while the amount of weak reactions is greater (77.3% vs 72.6%). Thus, our algorithm filters out more weak pupil size changes and therefore has lower recall value.

## 5.4 Qualitative subjective study

As an additional evaluation, we compare the recognition results of our algorithm with the expert opinions. For this purpose, we choose five more videos with the same visual parameters as the videos from Dataset A. These videos have been used neither for training the M-RCNN network nor for the parameters optimization or evaluation. We apply the proposed algorithm with configurations I and II to all of them. In Table 3, we present the number of detected reactions and the size of the largest detected pupil reaction for each video. The pupil reaction size

**Table 3. Results for the five new videos obtained with the configurations I and II.** For each video, the number of reactions per video and the size of the strongest detected reaction in each video are presented.

| Video | Detected reactions | | Strongest detected reaction | |
|---|---|---|---|---|
| | configuration I | configuration II | configuration I | configuration II |
| №1 | 6 | 4 | 12.26% | 13.56% |
| №2 | 10 | 3 | 10.55% | 8.64% |
| №3 | 14 | 17 | **17.10%** | 19.45% |
| №4 | 23 | 20 | **29.09%** | 26.79% |
| №5 | 16 | 17 | 13.98% | 15.52% |

**Table 4. The summary of expert opinions for the five new videos.** For each video, the frequency and the sizing of pupil reactions are presented.

| Video | Frequency | Comment about occurred reactions |
|---|---|---|
| №1 | rare | weak and medium |
| №2 | rare | the pupil becomes more and more narrow, but no dangerous reactions |
| №3 | often | medium and strong |
| №4 | very often | there are very strong and dangerous reactions |
| №5 | very often | unstable pupil, medium reactions |

is determined as follows: first, we compute a pupil to iris ratio at the beginning and the end of each pupil reaction. Then the difference between these values is considered as the size of the detected reaction. The size of the strongest pupil reaction in every processed video is shown in Table 3.

Also, we asked two medical experts to evaluate these videos in terms of the frequency and strength of pupil reactions. The experts' opinions are summarised in Table 4. Based on the prediction, it is noticeable that we can detect the videos that contain strong pupil reactions (№3, 4). Moreover, the frequency of the changes determined by experts correlates with the one provided by the automatic prediction algorithm.

## 6 Conclusion

In this work, we have evaluated automatic pupil reaction detection for moderately large datasets of videos from cataract surgery. Our approach uses object detection for pupil and limbus, and tracks the size of these objects over consecutive frames in the videos. It assumes that a significant size change of the pupil, together with a rather constant size of the limbus, indicates a pupil reaction of weak, moderate, or strong intensity (depending on the extent of the size change). For the evaluation we have manually annotated all pupil reactions in two video datasets, amounting to 30 cataract surgeries (about 650,000 frames), and compared these annotations to the segments proposed approach is able to automatically detect.

The results show that our method is able to accurately find pupil reactions of strong and medium intensity with a recall value up to 100%, meaning that all relevant segments are found. Although the precision values are rather low in this case, which means that also many wrong detections are contained in the results set, we argue that the proposed approach is still very helpful, as it allows to quickly filter candidate segments of pupil reactions, even though not all of them might be relevant (i.e., clinicians could quickly check the candidates, which is much faster than manually looking for pupil reactions). Here we would also like to stress that

many of the "wrong detections" in our evaluation are still valid from a technical point of view —i.e., most of them are slight pupil reactions, which are just not clinically relevant though. The evaluation results do also show that our method does not overfit but generalizes well to different data with other content characteristics.

Moreover, we compare the outcome of our algorithm with the experts' opinions. According to this evaluation, our method can detect how often and how strong the pupil reacts in cataract surgery videos similar to clinicians.

In the future work, we plan to improve the pupil reaction detection quality in cataract surgery videos using, for example, automatic machine learning approaches (such as AutoML). We also will extend the proposed algorithm with the ability to automatically categorize detected pupil reactions by the size/intensity. Using these automatic machine learning approaches, we are further planning to look at the clinical effect of intraoperative changes of the pupil size on the visual outcome of cataract surgeries. Pseudophakic cystoid macular edema (PCME), also called Irvine Gass Syndrome, is still a common cause of visual impairment after cataract surgery. Among various causes, blunt iris trauma has been attributed to play a major role in its development [29]. By applying the above-mentioned technique we do want to test this hypothesis. We are further planning to evaluate if the number of pupil variations or the amount of each pupil chance is the more relevant factor in the development of an Irvine Gass Syndrome. In conclusion, the development of an automatic evaluation of pupil changes will facilitate the evaluation of various settings in cataract surgeries and their role in their clinical outcome.

## Supporting information

**S1 Table.**
(PDF)

## Author Contributions

**Conceptualization:** Natalia Sokolova, Klaus Schoeffmann, Mario Taschwer, Stephanie Sarny, Doris Putzgruber-Adamitsch, Yosuf El-Shabrawi.

**Data curation:** Natalia Sokolova, Klaus Schoeffmann, Mario Taschwer, Doris Putzgruber-Adamitsch.

**Formal analysis:** Natalia Sokolova.

**Funding acquisition:** Klaus Schoeffmann, Yosuf El-Shabrawi.

**Investigation:** Natalia Sokolova.

**Methodology:** Natalia Sokolova, Mario Taschwer, Yosuf El-Shabrawi.

**Project administration:** Klaus Schoeffmann, Doris Putzgruber-Adamitsch, Yosuf El-Shabrawi.

**Resources:** Mario Taschwer.

**Software:** Natalia Sokolova, Klaus Schoeffmann, Mario Taschwer.

**Supervision:** Klaus Schoeffmann, Yosuf El-Shabrawi.

**Validation:** Natalia Sokolova, Doris Putzgruber-Adamitsch.

**Visualization:** Doris Putzgruber-Adamitsch.

**Writing – original draft:** Natalia Sokolova, Klaus Schoeffmann, Mario Taschwer.

Writing – review & editing: Klaus Schoeffmann, Yosuf El-Shabrawi.

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
