## [Decision Letter · Decision Letter 0]

2 Jul 2021

PONE-D-21-15823

Automatic detection of pupil reactions in cataract surgery videos

PLOS ONE

Dear Dr. El-Shabrawi,

Thank you for submitting your manuscript to PLOS ONE. After careful consideration, we feel that it has merit but does not fully meet PLOS ONE’s publication criteria as it currently stands. Therefore, we invite you to submit a revised version of the manuscript that addresses the points raised during the review process.

Reviewing details see below.

We look forward to receiving your revised manuscript.

Kind regards,

Andreas Wedrich

Academic Editor

PLOS ONE

Journal Requirements:

2. Thank you for including your ethics statement:  "The study was approved by the Local Ethic Board. (https://ethikkommission-kaernten.at)".   

Reviewers' comments:

Reviewer's Responses to Questions

**Comments to the Author**

1. Is the manuscript technically sound, and do the data support the conclusions?

Reviewer #1: Partly

Reviewer #2: Yes

2. Has the statistical analysis been performed appropriately and rigorously? 

Reviewer #1: I Don't Know

Reviewer #2: Yes

3. Have the authors made all data underlying the findings in their manuscript fully available?

Reviewer #1: Yes

Reviewer #2: Yes

4. Is the manuscript presented in an intelligible fashion and written in standard English?

Reviewer #1: Yes

Reviewer #2: Yes

5. Review Comments to the Author

Reviewer #1: In this study, the authors introduce an automatic method to detect pupil reaction in cataract surgery videos.

I do have some major concerns:

Validation of the outcome data is missing, i.e. human observers should be included. No sample size calculation is given and a number of 10 and 20, respectively, analysed patients should only be considered as a pre- study analysis. Given the low recall and precision values of this algorithm I suggest to re-evaluate the training data set. Auto ML deep learning might achieve much better recall and precision values.

Minor changes:

Recall, precision and GTCR should not be averaged in one value, i.e. harmonic mean, since they are highly divergent in this study.

Reviewer #2: This is an interesting and innovative manuscript on the automatic detection of pupil reactions in cataract surgery videos. Using Ai based on recorded videos may results in improved patient outcomes in the future.

The technical content of the manuscript is well described, but the flow of the manuscript may be improved.

1. Some parts of the manuscript dont read as scientific as they should. This is particularly true for the introduction and conclusion section. One example: "Clinicians need to operate very precisely to decrease the risk of intra- and postoperative complications to a minimum. Such issues are not only unpleasant for the patient but also costly for hospitals." I suggest that a senior clinican goes carefully through the manuscript and adapts a more scientific approach.

2. The conclusion section is summarizing the main results, but lacks the clinical relevance of the approach. Again, a clinician has to take the lead on this part and provide a future persepctive for catarct surgery,

3. The authors need to help clinical readers to better understand their approach. Table 1 is given without much explanation on what these different measures (H, Recall, ...) mean.

4. The paper needs to be corretced by a native speaker. Some of the sentences are very difficult to understand. Examples include "As unclear changes are difficult to detect even with the human eye, we assume that the results should be better if we exclude such pupil size changes from our evaluation." "To keep the numbering of the frames, we insert the missing 176 values with the last satisfying pupil and iris width values." "In this work, we operate with the following terms: pupil, iris, and limbus (a line between the iris and the outer part of the eye)."

6. PLOS authors have the option to publish the peer review history of their article (what does this mean?). If published, this will include your full peer review and any attached files.

Reviewer #1: No

Reviewer #2: No

---

## [Author Response · Author response to Decision Letter 0]

8 Sep 2021

We thank the editor and the reviewers for their valuable feedback that helped us to further improve our paper. We have significantly revised our manuscript and tried to resolve all mentioned concerns, as we describe down below. In summary, we have performed the following changes:

- Ethics statement information;

- Improved description of used materials (Sections 2.1 and 2.2);

- Extended evaluation results with the detailed analysis for pupil reactions of different size groups (Section 4.2.3);

- Qualitative subjective study (Section 4.4);

- Extended future work (Conclusion);

- Improved presentation and writing throughout the paper.

To highlight the changes in our current manuscript, we use different colors for the editor and the different reviewers. The colors are assigned as follows: pink for the answers to the editor, green for the responses to reviewer one, and yellow for the responses to reviewer two.

Please find our detailed responses to the comments in the rebuttal letter attached to the current submission.

We also would like to ask the dear editor to update the Financial Disclosure information of our submission to the following:

"This work was funded by the FWF Austrian Science Fund under grant P 31486-N31.".

---

## [Decision Letter · Decision Letter 1]

27 Sep 2021

Automatic detection of pupil reactions in cataract surgery videos

PONE-D-21-15823R1

Dear Dr. El-Shabrawi,

We’re pleased to inform you that your manuscript has been judged scientifically suitable for publication and will be formally accepted for publication once it meets all outstanding technical requirements.

Kind regards,

Andreas Wedrich

Academic Editor

PLOS ONE

Additional Editor Comments (optional):

Reviewers' comments:

Reviewer's Responses to Questions

**Comments to the Author**

1. If the authors have adequately addressed your comments raised in a previous round of review and you feel that this manuscript is now acceptable for publication, you may indicate that here to bypass the “Comments to the Author” section, enter your conflict of interest statement in the “Confidential to Editor” section, and submit your "Accept" recommendation.

Reviewer #2: All comments have been addressed

2. Is the manuscript technically sound, and do the data support the conclusions?

Reviewer #2: Yes

3. Has the statistical analysis been performed appropriately and rigorously? 

Reviewer #2: Yes

4. Have the authors made all data underlying the findings in their manuscript fully available?

Reviewer #2: Yes

5. Is the manuscript presented in an intelligible fashion and written in standard English?

Reviewer #2: Yes

6. Review Comments to the Author

Reviewer #2: All issues addressed, this is an intresting stuy that merits publication. The results ar clearly persented.

7. PLOS authors have the option to publish the peer review history of their article (what does this mean?). If published, this will include your full peer review and any attached files.

Reviewer #2: No

---

## [Editor Report · Acceptance letter]

1 Oct 2021

PONE-D-21-15823R1 

Automatic detection of pupil reactions in cataract surgery videos 

Dear Dr. El-Shabrawi:

I'm pleased to inform you that your manuscript has been deemed suitable for publication in PLOS ONE. Congratulations! Your manuscript is now with our production department. 

Kind regards, 

on behalf of

Prof. Dr. Andreas Wedrich 

Academic Editor

PLOS ONE